# Quality of life, healthcare use and costs in 'at-risk' children after early antibiotic treatment versus placebo for influenza-like illness: within-trial descriptive economic analyses of the ARCHIE randomised controlled trial

Ines Rombach ⓘ ,[1] Kay Wang ⓘ ,[2] Sharon Tonner ⓘ ,[2] Jenna Grabey ⓘ ,[2] Anthony Harnden ⓘ ,[2] Jane Wolstenholme ⓘ ,[1] For the ARCHIE Collaborators Group

[1]Health Economics Research Centre, Nuffield Department of Population Health, University of Oxford, Oxford, UK
[2]Nuffield Department of Primary Care Health Sciences, University of Oxford, Oxford, UK

**Correspondence to**
Dr Jane Wolstenholme;
jane.wolstenholme@ndph.ox.ac.uk

## ABSTRACT

**Objectives** To characterise the quality of life, healthcare use and costs associated with early antibiotic treatment of influenza-like illness (ILI) in 'at-risk' children.

**Design** Economic analysis of a two-arm double-blind parallel group pragmatic randomised controlled trial.

**Setting** Children were recruited from community-based healthcare settings, including general practices, walk-in centres and hospital ambulatory care.

**Participants** Children with risk factors for influenza-related complications, including respiratory, cardiac and neurological conditions, who presented within the first 5 days of an ILI.

**Interventions** Co-amoxiclav 400/57 suspension or placebo.

**Outcome measures** This economic analysis focused on quality of life measured by the EQ-5D-Y, symptoms assessed by the Canadian Acute Respiratory Infection and Flu Scale (CARIFS), healthcare use and costs including medication, hospital visits and admissions, general practitioner and nurse contacts. Outcomes were assessed for up to 28 days post randomisation.

**Results** Information on resource use, EQ-5D-Y (day 28) and CARIFS (day 7) was available for 265 (98%), 72 (27%) and 123 (45%) out of 271 participants, respectively. Average costs in the co-amoxiclav group were £25 lower (95% CI −£113 to £65), but this difference was not statistically significant (p=0.566). The difference in EQ-5D-Y scores between groups was also not statistically significant (−0.014 (95% CI −0.124 to 0.096), p=0.798). However, day 7 CARIFS scores were 3.5 points lower in the co-amoxiclav arm (95% CI −6.9 to −0.1, p=0.044).

**Conclusions** Our findings did not show evidence that early co-amoxiclav treatment improves quality of life or reduces healthcare use and costs in 'at-risk' children with ILI, but may reduce symptom severity though confirmation from further research would be important. Reliable data collection from children's parents/carers was challenging, and resulted in high levels of missing data, which is common in pragmatic trials involving children with acute respiratory tract infections.

## Strengths and limitations of this study

- ► This study aimed to describe quality of life, healthcare use and costs in children with influenza-like illness (ILI) and known risk factors for complications.
- ► The study used pragmatic inclusion criteria, recruiting children with ILI, which reflects current clinical practice but cannot reliably distinguish between influenza infection and ILI due to other infections.
- ► Subgroup analyses examined differential costs between children with and without laboratory-confirmed influenza.
- ► Reviews of medical records provided robust information on healthcare use with negligible levels of missing data.
- ► Given the nature of the participant group, proxy-reported completion of instruments and questionnaires was required to inform the quality of life, healthcare use and costs, however, this was subject to high levels of missing data, making a formal cost-effectiveness analysis impossible to undertake.

**Trial registration number** ISRCTN70714783; EudraCT 2013-002822-21.

## BACKGROUND

The clinical results of the 'Early use of Antibiotics for at Risk CHildren with InfluEnza-like illness' (ARCHIE) trial were recently published.[1] This randomised controlled trial (RCT) aimed to assess uncertainty in the treatment of influenza-like illness (ILI) in 'at-risk' children, that is, children with pre-existing conditions such as asthma, diabetes mellitus and cerebral palsy.

The RCT compared early treatment with oral co-amoxiclav against placebo, and used proportion of children reconsulting due to clinical deterioration within 28 days of randomisation as the primary outcome measure.

Full details of the trial are published elsewhere.[1 2] The primary intention-to-treat analysis did not show a statistically significant difference between the co-amoxiclav and placebo groups, as measured by the proportion of children reconsulting due to clinical deterioration within 28 days. A reconsultation rate of 24.8% was observed in the co-amoxiclav arm, and 21.2% in the placebo arm (adjusted risk ratio 1.16 (95% CI 0.75 to 1.80). An exploratory analysis in the subgroup with laboratory-confirmed influenza showed that a lower proportion of children in the co-amoxiclav group reconsulted due to clinical deterioration, although this subgroup did not have sufficient power to demonstrate whether this difference was statistically significant.

The trial also investigated health-related quality of life (HRQoL), healthcare use and cost associated with the early co-amoxiclav use in 'at-risk' children. This research is important due to the absence of QoL and healthcare use data associated with early antibiotic use associated with influenza or ILI in this important patient population. Our literature review for our pre-trial modelling work demonstrated a lack of observational data in these 'at-risk' children, and we were unable to identify any clinical trials in this paediatric cohort.[3–5] Previous UK based cost-effectiveness models of antivirals for treatment of influenza in children used adult utility values based on derived values from adults treated with oseltamivir.[6 7] An alternative source of child-specific proxy EQ-5D data was from a trial of corticosteroids for otitis media.[8]

## METHODS

Design and planned analyses for the ARCHIE trial, including the health economic analysis, are described in the published protocol.[2] Specific analyses of the QoL, healthcare use and cost data are described below. Data were collected for a UK-based National Health Service (NHS) perspective, with additional data collected to provide a wider societal perspective. The time horizon of this study is 28 days.

### Study population

The ARCHIE trial recruited 271 children at increased risk of influenza-related clinical deterioration or complications. Recruited children were between the ages of 6 months to 12 years and within the first 5 days of an ILI. In the remit of this pragmatic trial, ILI was defined as the presence of both cough and fever, whereby fever could either be child-reported, parent-reported or temperature >37.8°C (axillary or tympanic temperature measurement). 'At-risk' was based on healthcare professionals' clinical judgement and included children with pre-existing conditions such as asthma, diabetes mellitus and cerebral palsy. Children were ineligible if they had known

contraindications to co-amoxiclav, required immediate antibiotics based on clinical judgement, required immediate hospital admission for an influenza-related complication (clinical judgement), or had been observed on a hospital ward or ambulatory care unit for longer than 24 hours. Full details of all inclusion and exclusion criteria are given in the trial protocol.[9]

Participants were recruited from general practices, walk-in centres and hospitals in the UK with a view to making healthcare decisions in the NHS and other healthcare settings.

The planned sample size for the trial was 650, based on an absolute risk reduction in reconsultations due to clinical deterioration of 14% (reduction from 40% to 26%), using 90% power, a two-sided 5% significance level, an inflation factor of 1.041, and up to 25% loss to follow-up. The full sample size was not reached, and recruitment was closed after 271 children had been randomised.

Children were randomised 1:1 to either co-amoxiclav 400/57 (amoxicillin 400 mg as trihydrate and clavulanic acid 57 mg as potassium salt/5 mL when reconstituted with water) or a matching placebo, to be taken orally twice daily for 5 days. Co-amoxiclav was used due to its susceptibility in treating lower respiratory tract bacterial isolates associated with influenza in primary care,[10] and is stockpiled by the UK government for use in influenza pandemics; a matching placebo was chosen to obtain unbiased results for treatment effects.[1]

### Quality of life

QoL was measured using the EuroQoL 5-Dimension 3-Level (EQ-5D 3-L) youth version (EQ-5D-Y), which was confirmed in pre-trial focus groups as most applicable in this patient population, compared with other questionnaires (ie, Child Health Utility-9D).[11] The EQ-5D-Y was collected at baseline, days 4, 7, 14 and 28. For all children, parents or carers (ie, proxies) completed the data on the children's behalf. Data were also collected from the participants directly, where appropriate. No restrictions on which children should complete the EQ-5D-Y themselves were included in the protocol. This was due to some of the participants having complex needs and therefore being unable to complete questionnaires regardless of their age. Equally, severe illness may have prevented some children who would usually be able to complete the data from doing so at some trial follow-up time points. Whether or not children were asked to complete questionnaires was left to their proxy's discretion.

The EQ-5D-Y consists of five questions covering mobility, self-care, usual activities, pain/ discomfort, and anxiety/ depression. Each question has three response levels; 'no', 'some', 'a lot'. As the scoring for the youth-version had not been published at the time of the analysis of this study,[11 12] the responses to the five questions were converted into utilities using the EQ-5D-3L algorithm for the UK value set,[13–15] ranging from −0.594 to 1, where 1 indicates perfect health.

The health state utilities calculated from the responses to EQ-5D-Y were used to estimate mean quality-adjusted life years (QALYs) rated on behalf of the child and by the child for both trial arms at 28 days. This was done using an area under the curve approach, whereby utilities were assumed to change linearly between each follow-up time point, weighted by the time periods the questionnaires were administered, that is, days 0 (baseline), 4, 7, 14 and 28. QALYs were generated only for participants who had EQ-5D-Y data available at baseline and all follow-up time points to 28 days.

### Canadian Acute Respiratory Infection and Flu Scale

The Canadian Acute Respiratory Infection and Flu Scale (CARIFS)[16] consists of 18 questions covering 3 domains, that is, symptoms, function and parental impact. The questions have 4-point ordinal scale, from no problems to major problems, scored 0 to 3. A total score ranging from 0 (best possible health) to 54 (worst possible health) is calculated by adding up all 18 items. Missing data in the three questions about headache, sore throat, muscle aches or pains were handled by mean imputation, in line with the scoring instructions.[16] The CARIFS was collected at baseline and 1 week.

### Healthcare use

Proxy-reported healthcare use was collected in the study diaries with regards to medication use, hospital visits and admissions, general practitioner (GP) and nurse contacts. Information on the number of days children were unable to attend school or nursery, and subsequent changes to childcare requirements were also collected. These data were collected in weekly diaries on days 7, 14, 21 and 28, requiring recall of healthcare resource use over the previous week.

A medical notes review was conducted to obtain clinical data for the primary outcome (reconsultation due to clinical deterioration), medication prescriptions and/or further investigations, and hospitalisations or deaths within 28 days of randomisation. The number of reconsultations, antibiotics and other drugs given, chest X-rays and other investigations performed were included in this review and were used in the estimation of healthcare use.

For the purposes of this analysis, only reconsultations due to clinical deterioration considered as related to the initial illness episode for which the child was randomised were included, in line with the primary endpoint of the trial. Reconsultations had to take place within 28 days from randomisation and in relation to worsening symptoms, new symptoms or complications for which the child required medication, investigation or referral to hospital.

Referrals to hospital or emergency departments for acute admission were recorded from the medical notes review including information on hospital inpatient admissions, medications and investigations undertaken.

The study's clinical lead reviewed all hospital admissions, and distinguished between those considered to be related to the illness episode the child was randomised

for, and those not related. They also reviewed all medication recorded. Medications deemed unrelated to the illness for which the child was recruited into the trial were identified and excluded from this analysis.

Information on healthcare use was restricted to the information obtained via the medical notes review. Information from the carer-completed diaries was considered insufficient for inclusion in this analysis (details on data availability for the self-reported questionnaires are provided in the results section for information). Serious adverse events as reported by the recruiting sites were reviewed to ensure all relevant healthcare use was captured through the medical notes review.

### Unit costs

Costs were for the UK NHS perspective.

Participating children were recruited from GP practices and hospital-based ambulatory assessment units. Unit costs for GP practice visits or reconsultations with a GP were obtained from the Personal Social Services Research Unit (PSSRU) Unit Costs of Health and Social Care compendium for 2017/2018.[17] Unit costs for all other health service contacts were derived from the NHS National Schedule of Reference Costs 2017–2018,[18] including reconsultations for participants recruited in paediatric ambulatory assessment units, which were costed as a visit to a paediatric ambulatory assessment unit.

Investigations, including blood tests, nasal aspirate and throat swabs were costed based on information on Directly Accessed Pathology Services in the NHS National Schedule of Reference Costs 2017–2018.[18] Costs of urine tests were based on information published by Fenwick et al.[19]

Antibiotics and other prescription drug costs were based on the British National Formulary (BNF for children) and UK tariff costs,[20] assuming the minimum value reported. Where possible, unit costs were applied based on the dose reported, or a median dose if no dose was reported; cost for inhalers and eye/nose drops were based on the minimum unit sold.

Information on the unit costs included in this trial is presented in online supplemental file 1.

Due to the 28-day time horizon, no discounting was used.

### Analysis

#### Within-trial analysis

Summaries were generated on an as randomised basis, with participants analysed within their randomised groups regardless of compliance with their allocated interventions. All available data were used, only participants whose parents or carers withdrew consent for extraction of data from their children's medical record, or where access to medical notes was not possible for other reasons were excluded from the summaries. No imputations were performed for missing data.

Data availability was summarised for the EQ-5D-Y (completed by the child, and on their behalf (proxy)), CARIFS, availability of daily activity and childcare questionnaire, and availability of health service contact (proxy-reported).

EQ-5D-Y, QALYs and CARIFS data were summarised descriptively by data collection time point. Healthcare use was summarised using number of events and frequencies to indicate how often participants had used each service. Mean number of contacts with different health services were also calculated. Costs incurred for the different health service contacts were calculated by applying the unit costs, described in online supplemental file 1. Information on the number of days children were unable to attend school or nursery, and subsequent changes to childcare requirements reported during each of the follow-up weeks were also summarised.

Differences between trial arms were estimated using linear regression models, adjusted for the minimisation factors; age (used as continuous variable) and seasonal influenza vaccination status. Clustering by the recruiting site (ie, the GP practices or hospital ambulatory units at which participants were randomised) was accounted for using the 'cluster' option in Stata's 'regress' command and robust standard errors were generated.

### Additional analyses and sensitivity analyses

Subgroup analyses explored if treatment effects varied depending on whether the children's illness at baseline was laboratory-confirmed to be influenza, by adding an interaction term between treatment allocation and laboratory-confirmed influenza at baseline to the above described analysis model. Subgroup analyses were performed for the total costs incurred over the trial follow-up period, and separately for the costs incurred through medication use, and the costs related to reconsultations, hospital admissions and referrals. Results from this investigation were presented as forest plots. As this analysis was exploratory, no p values were displayed.

Sensitivity analysis examined the effect of including costs associated with hospitalisations unrelated to the illness episode for which the child was recruited to ARCHIE.

### Extrapolation of findings to national scale

The costs were extrapolated to estimate the impact of either intervention on a national scale. The number of children within the relevant age category in the UK were estimated from information for 2018 from the Office for National Statistics.[21] The number of GP consultations during 2018/2019 was estimated from the 'Surveillance of influenza and other respiratory viruses in the UK'.[22] Finally, the proportions of children who consult with GPs for ILI who correspond to the ARCHIE trial population were estimated from the current literature.[9]

From these figures, the total number of 'at-risk' children likely to consult a GP with ILI was estimated for the UK. This figure was combined with the estimations of

cost differences between the co-amoxiclav and placebo groups, to estimate the likely monetary impact of the co-amoxiclav intervention on a national scale.

### Patient and public involvement

The research team conducted consultations with parents of 'at-risk' children and young people from the National Institute for Health Research Generation R Young Person's Advisory Group to inform the design and conduct of the ARCHIE programme. The ARCHIE Programme Steering Committee also included a patient representative.

## RESULTS

### Data availability

In total, 271 children were randomised into the trial, 136 to co-amoxiclav, 135 to placebo. Details of the study population characteristics are provided in table 1. Six children were excluded. Of these, five children were withdrawn from the trial and all follow-up, and medical notes of one child could not be accessed. As such, 265 children were included in the analysis (133 randomised to co-amoxiclav, 132 to placebo). All 265 children received their allocated intervention, and there were no cross-overs between the treatment arms. Data on healthcare resource use based on note reviews were available for all 265 children.

Proxy-completed EQ-5D-Y data were available for 75% (204/271) of children at baseline, and 27% (72/271) at 28 days. Self-completed EQ-5D-Y data were available for 23% (61/271) at baseline, and 10% (27/271) at 28 days. CARIFS data were completed for 72% (196/271) of children at baseline and 45% (123/271) at day 7. Proxy-reported data for daily activity and childcare questionnaire, and the health service contact questions were available for 54% (146/271) at baseline, and less than 35% at 28 days respectively (92/271 daily activity and childcare questionnaires and 87/271 healthcare use questionnaires were received). Table 2 shows this information by treatment arm, and for all follow-up points. Additional information on data availability, including by age group, and EQ-5D-Y missing data pattern over time are available in online supplemental file 2.

### QoL and healthcare use

Data for the EQ-5D-Y, QALYs and CARIFS are shown in table 3. There were no statistically significant differences between the treatment arms for the EQ-5D-Y and QALYs. The EQ-5D VAS completed on behalf of the children was statistically significantly higher, indicating higher QoL, in the co-amoxiclav arm at 7 days postrandomisation only. The CARIFS score was significantly lower in the co-amoxiclav arm, indicating lower severity of respiratory symptoms in those children. Reponses to the individual EQ-5D-Y and CARIFS items, for the whole cohort and split by age group, are presented in online supplemental file 3.

Details of healthcare use for related reconsultations due to clinical deterioration, and hospital admissions related to the illness episode for which the child was randomised

**Table 1** Characteristics of the study population

| | Co-amoxiclav (N=136) | Placebo (N=135) | Total (N=271) |
|---|---|---|---|
| Age in months* | 40.8 (19.4, 85.6) | 36.4 (20.9, 70.8) | 39.3 (20.2, 78.2) |
| Female† | 53 (39%) | 55 (41%) | 108 (40%) |
| At least one acute consultation in the 12 month period before entering the study† | 123 (90%) | 119 (88%) | 242 (89%) |
| Antibiotics prescribed in preceding 3 months† | 33 (23%) | 25 (19%) | 58 (21%) |
| 'At-risk' category (not mutually exclusive)† | | | |
| Respiratory | 99 (73%) | 99 (73%) | 198 (73%) |
| Neurological | 6 (4%) | 9 (7%) | 15 (6%) |
| Cardiac | 12 (9%) | 4 (3%) | 16 (6%) |
| Renal | 3 (2%) | 0 (0%) | 3 (1%) |
| Immunodeficiency | 1 (1%) | 0 (0%) | 1 (0%) |
| Genetic | 9 (7%) | 9 (7%) | 18 (7%) |
| Metabolic | 1 (1%) | 5 (4%) | 6 (2%) |
| Premature birth | 13 (10%) | 15 (11%) | 28 (10%) |
| Previous recurrent or serious respiratory problems | 6 (4%) | 8 (6%) | 14 (5%) |
| Other (allergies, Hx astro astana) | 3 (2%) | 3 (2%) | 6 (2%) |
| Duration of illness (days)* | 2.7 (1.2) | 2.7 (1.2) | 2.7 (1.2) |

Denominators are given for variables with missing data.
*Median (IQR) or mean (SD).
†Frequency (%).

into ARCHIE are shown in table 4. Overall, 23% of children (61/265) had at least one reconsultation due to clinical deterioration, and hospital admissions related to the illness episode for which the child was randomised into ARCHIE were reported for 5% of children (12/265). Additional information on all hospital admissions and average healthcare use, as typically presented in cost-effectiveness analyses, are shown in online supplemental file 4. Resource use was similar between the treatment arms.

Costs per participant associated with related reconsultations and hospital admissions are shown in table 5. Average total costs over 28-day follow-up period were estimated as £82 (SD 481) in the co-amoxiclav arm, and £117 (SD 540) in the placebo arm. The medians and IQRs of zero for most costing items reflect the fact that 75% of the participants did not use the relevant resources and hence did not incur corresponding costs. There were no statistically significant differences in total costs, or any of the cost-components, between the treatment groups. Hospital stays appear to be the main cost drivers.

Information on the number of days children were unable to attend school or nursery, and subsequent changes to childcare requirements are summarised in online supplemental file 5. There were no statistically significant differences between the groups.

### Subgroup analysis
Figure 1 shows the difference in costs between the treatment arms by whether or not influenza was confirmed through nasal swabs taken at baseline. Two hundred and sixty-two children for whom the results of nasal swabs were available are included in these summaries. Of these, 21 participants in the co-amoxiclav group (15%), and 16 participants in the placebo group (12%) had laboratory-confirmed influenza. There was a trend towards lower costs in the co-amoxiclav arm for children with laboratory-confirmed influenza, though these differences were not statistically significant.

### Extrapolation of findings to national scale
According to the Office for National Statistics, in mid-2018,[21] there were approximately 9.7 million children aged 6 months to 12 years in the UK. Information from the 'Surveillance of influenza and other respiratory viruses in the UK' indicates that maximum weekly rates of GP consultations for ILI in 2018/2019 were around 21 per 100 000 for children aged <1 years, 25 per 100 000 for children aged 1–4 years and 19 per 100 000 for children aged 5–14.[22]

We assume approximately 21 per 100 000 GP consultations were for influenza-like symptoms over an 8 week period in children. Weekly GP consultations for the remaining 44 weeks of the year were assumed to be 5 per 100 000.

Finally, Lee et al[9] estimate that 15.3% of children with influenza like symptoms suffer from at least one of the underlying conditions that were used to classify children as 'at-risk', and hence eligible for the ARCHIE trial.

**Table 2** Overview of availability for self-reported data

| | Co-amoxiclav (N=136) | Placebo (N=135) | Total (N=271) |
|---|---|---|---|
| Availability of EQ-5D-Y index completed on behalf of child | | | |
| Day 0 | 96 (71%) | 108 (80%) | 204 (75%) |
| Day 4 | 52 (38%) | 52 (39%) | 104 (38%) |
| Day 7 | 52 (38%) | 54 (40%) | 106 (39%) |
| Day 14 | 44 (32%) | 46 (34%) | 90 (33%) |
| Day 28 | 41 (30%) | 31 (23%) | 72 (27%) |
| Availability of EQ-5D VAS completed on behalf child | | | |
| Day 0 | 120 (88%) | 129 (96%) | 249 (92%) |
| Day 4 | 77 (57%) | 76 (56%) | 153 (56%) |
| Day 7 | 78 (57%) | 73 (54%) | 151 (56%) |
| Day 14 | 61 (45%) | 57 (42%) | 118 (44%) |
| Day 28 | 53 (39%) | 40 (30%) | 93 (34%) |
| Availability of EQ-5D-Y index completed by child | | | |
| Day 0 | 32 (24%) | 29 (21%) | 61 (23%) |
| Day 4 | 16 (12%) | 13 (10%) | 29 (11%) |
| Day 7 | 17 (13%) | 11 (8%) | 28 (10%) |
| Day 14 | 17 (13%) | 10 (7%) | 27 (10%) |
| Day 28 | 17 (13%) | 10 (7%) | 27 (10%) |
| Availability of EQ-5D VAS completed by child | | | |
| Day 0 | 28 (21%) | 25 (19%) | 53 (20%) |
| Day 4 | 19 (14%) | 13 (10%) | 32 (12%) |
| Day 7 | 19 (14%) | 11 (8%) | 30 (11%) |
| Day 14 | 20 (15%) | 10 (7%) | 30 (11%) |
| Day 28 | 18 (13%) | 11 (8%) | 29 (11%) |
| Availability of CARIFS | | | |
| Day 0 | 99 (73%) | 97 (72%) | 196 (72%) |
| Day 7 | 66 (49%) | 57 (42%) | 123 (45%) |
| Availability of CARIFS VAS | | | |
| Day 0 | 133 (98%) | 135 (100%) | 268 (99%) |
| Day 7 | 76 (56%) | 74 (55%) | 150 (55%) |
| Availability of daily activity and childcare questionnaire | | | |
| Day 7 | 75 (55%) | 71 (53%) | 146 (54%) |
| Day 14 | 61 (45%) | 57 (42%) | 118 (44%) |
| Day 21 | 54 (40%) | 46 (34%) | 100 (37%) |
| Day 28 | 53 (39%) | 39 (29%) | 92 (34%) |
| Availability of healthcare use questionnaire | | | |
| Day 7 | 76 (56%) | 70 (52%) | 146 (54%) |
| Day 14 | 56 (41%) | 52 (39%) | 108 (40%) |
| Day 21 | 51 (38%) | 45 (33%) | 96 (35%) |
| Day 21 | 51 (38%) | 45 (33%) | 96 (35%) |
| Day 28 | 49 (36%) | 38 (28%) | 87 (32%) |

EQ-5D-Y and CARIFS were included as 'available' if all relevant questions were completed; the daily activities and childcare questionnaire, and the health service contact questionnaire were considered 'available' if at least one question had been answered, including indicating that the daily activities and childcare questions were not applicable.
CARIFS, Canadian Acute Respiratory Infection and Flu Scale; EQ-5D, EuroQoL 5-Dimension; EQ-5D-Y, EuroQoL 5-Dimension 3-Level youth version; VAS, Visual Analogue Scale.

**Table 3** EQ-5D-Y, quality adjusted life scores and CARIF outcomes

| | Co-amoxiclav | | Placebo | | Difference | |
|---|---|---|---|---|---|---|
| | N | Mean (SD) | N | Mean (SD) | Total N | Difference (95% CI)* |
| **EQ-5D-Y proxy, completed on behalf of child** | | | | | | |
| Day 0 | 96 | 0.591 (0.342) | 108 | 0.568 (0.386) | | |
| Day 4 | 52 | 0.630 (0.349) | 52 | 0.690 (0.318) | 104 | −0.057 (95% CI −0.198 to 0.085), p=0.425 |
| Day 7 | 52 | 0.788 (0.360) | 54 | 0.762 (0.343) | 106 | 0.037 (95% CI −0.102 to 0.176), p=0.596 |
| Day 14 | 44 | 0.841 (0.338) | 46 | 0.923 (0.157) | 90 | −0.086 (95% CI −0.200 to 0.028), p=0.135 |
| Day 28 | 41 | 0.905 (0.246) | 31 | 0.923 (0.169) | 72 | −0.014 (95% CI −0.124 to 0.096), p=0.798 |
| **EQ-5D VAS completed on behalf of child** | | | | | | |
| Day 0 | 120 | 60 (17) | 129 | 58 (19) | | |
| Day 4 | 77 | 72 (18) | 76 | 68 (20) | 153 | 4.1 (95% CI −1.5 to 9.7), p=0.146 |
| Day 7 | 78 | 83 (18) | 73 | 76 (22) | 151 | 7.9 (95% CI 1.3 to 14.4), p=0.019 |
| Day 14 | 61 | 87 (19) | 57 | 90 (13) | 118 | −3.7 (95% CI −8.7 to 1.3), p=0.139 |
| Day 28 | 53 | 89 (17) | 40 | 86 (21) | 93 | 4.1 (95% CI −5.0 to 13.3), p=0.368 |
| **EQ-5D-Y completed by child** | | | | | | |
| Day 0 | 32 | 0.469 (0.388) | 29 | 0.531 (0.363) | | |
| Day 4 | 16 | 0.626 (0.386) | 13 | 0.717 (0.227) | 29 | −0.107 (95% CI −0.372 to 0.157), p=0.408 |
| Day 7 | 17 | 0.815 (0.386) | 11 | 0.805 (0.180) | 28 | −0.034 (95% CI −0.336 to 0.269), p=0.819 |
| Day 14 | 17 | 0.957 (0.086) | 10 | 0.918 (0.144) | 27 | 0.028 (95% CI −0.073 to 0.130), p=0.565 |
| Day 28 | 17 | 0.986 (0.039) | 10 | 0.941 (0.124) | 27 | 0.035 (95% CI −0.039 to 0.109), p=0.330 |
| **EQ-5D VAS completed by child** | | | | | | |
| Day 0 | 28 | 54 (22) | 25 | 55 (21) | | |
| Day 4 | 19 | 62 (22) | 13 | 59 (21) | 32 | 4.2 (95% CI −11.2 to 19.7), p=0.577 |
| Day 7 | 19 | 73 (21) | 11 | 73 (25) | 30 | −0.7 (95% CI −24.9 to 23.4), p=0.952 |
| Day 14 | 20 | 84 (19) | 10 | 83 (26) | 30 | 1.4 (95% CI −19.8 to 22.6), p=0.893 |
| Day 28 | 18 | 88 (21) | 11 | 84 (25) | 29 | 3.0 (95% CI −9.2 to 15.2), p=0.612 |
| QALYs—rated on behalf of child | 29 | 0.061 (0.022) | 25 | 0.065 (0.011) | 54 | −0.003 (95% CI −0.012 to 0.007), p=0.573 |
| QALYs—rated by child | 9 | 0.069 (0.004) | 4 | 0.065 (0.007) | 13 | 0.003 (95% CI −0.004 to 0.009), p=0.357 |
| **CARIFS** | | | | | | |
| Day 0 | 99 | 22 (10) | 97 | 23 (11) | | |
| Day 7 | 66 | 8 (8) | 57 | 12 (11) | 123 | −3.5 (95% CI −6.9 to 0.1), p=0.044 |
| **CARIFS VAS** | | | | | | |
| Day 0 | 133 | 4.64 (1.90) | 135 | 4.78 (1.92) | | |
| Day 7 | 76 | 1.91 (1.97) | 74 | 2.53 (2.45) | 150 | −0.63 (95% CI −1.38 to 0.11), p=0.096 |

EQ-5D-Y utilities: range −0.594 to 1, where 1 indicates perfect health.
EQ-5D VAS: range 0–100, where higher values indicate better health.
QALYs were calculated only for children with data available at all relevant time points.
CARIFS: range 0–54 points, with lower scores indicating fewer respiratory illness and influenza symptoms.
CARIFS VAS: range 0–10, with higher scores indicating worse health states.
*Differences have been adjusted for the stratification factors age (used as continuous variable) and seasonal influence vaccination status. Clustering by centre has been accounted for using the 'cluster' option in Stata's 'regress' command and robust standard errors were generated.
CARIFS, Canadian Acute Respiratory Infection and Flu Scale; EQ-5D-Y, EuroQoL 5-Dimension 3-Level youth version; QALYs, quality-adjusted life-years; VAS, Visual Analogue Scale.

This means that UK-wide, there were around 37 821 GP consultations for influenza like symptoms, 15.3% (5787) of which would have been expected to occur in the 'at-risk' groups included in the ARCHIE study. Using the estimated cost difference between the treatment arms of −£25 (95% CI −£113 to £62), the potential cost saving of early co-amoxiclav treatment compared with not providing early antibiotics is estimated as £−144 675 (95% CI £−358 794 to £653 931).

## DISCUSSION

In this study, we have summarised the HRQoL, healthcare use and costs associated with treating 'at-risk' children with ILI with co-amoxiclav versus placebo.

**Table 4** Details of healthcare use for related reconsultations due to clinical deterioration and hospital admissions related to the illness episode for which the child was randomised into ARCHIE over the 28-day follow-up period

| | Co-amoxiclav (N=133) | Placebo (N=132) | Total (N=265) |
|---|---|---|---|
| Related reconsultations | | | |
| No of participants with at least one related reconsultation due to clinical deterioration | 33 (25%) | 28 (21%) | 61 (23%) |
| No of related reconsultations due to clinical deterioration | | | |
| 0 | 100 (75%) | 104 (79%) | 204 (77%) |
| 1 | 30 (23%) | 25 (19%) | 55 (21%) |
| 2 | 3 (2%) | 3 (2%) | 6 (2%) |
| No of participants with at least one related reconsultation due to clinical deterioration in a primary care or equivalent ambulatory care setting (excluding hospital admissions)* | 28 (21%) | 23 (17%) | 51 (19%) |
| Related reconsultation due to clinical deterioration in a primary care or equivalent ambulatory care setting (excluding hospital admissions)* | | | |
| 0 | 105 (79%) | 109 (83%) | 214 (81%) |
| 1 | 25 (19%) | 21 (16%) | 46 (17%) |
| 2 | 3 (2%) | 2 (2%) | 5 (2%) |
| No of antibiotics received at related reconsultation* | | | |
| 0 | 120 (90%) | 119 (90%) | 239 (90%) |
| 1 | 13 (10%) | 13 (10%) | 26 (10%) |
| No of other drugs received at related reconsultation* | | | |
| 0 | 126 (95%) | 127 (96%) | 253 (95%) |
| 1 | 6 (5%) | 1 (1%) | 7 (3%) |
| 2 | 1 (1%) | 1 (1%) | 2 (1%) |
| 3 | 0 (0%) | 3 (2%) | 3 (1%) |
| No of chest X-rays at related reconsultation* | | | |
| 0 | 129 (97%) | 131 (99%) | 260 (98%) |
| 1 | 4 (3%) | 1 (1%) | 5 (2%) |
| No of other interventions at related reconsultation*‡‡ | | | |
| 0 | 131 (98%) | 132 (100%) | 263 (99%) |
| 1 | 2 (2%) | 0 (0%) | 2 (1%) |
| Related hospital admissions | | | |
| No of hospital admissions | | | |
| 0 | 128 (96%) | 125 (95%) | 253 (95%) |
| 1 | 5 (4%) | 7 (5%) | 12 (5%) |
| Total nights in hospital | | | |
| 0 | 128 (96%) | 125 (95%) | 253 (95%) |
| 1 | 3 (2%) | 1 (1%) | 4 (2%) |
| 2 | 1 (1%) | 4 (3%) | 5 (2%) |
| 3 | 0 (0%) | 1 (1%) | 1 (0%) |
| 7 | 1 (1%) | 1 (1%) | 2 (1%) |
| No of antibiotics received during hospital admission | | | |
| 0 | 131 (98%) | 126 (95%) | 257 (97%) |
| 1 | 2 (2%) | 6 (5%) | 8 (3%) |
| No of other drugs received during hospital admission | | | |
| 0 | 129 (97%) | 126 (95%) | 255 (96%) |
| 1 | 1 (1%) | 0 (0%) | 1 (0%) |
| 2 | 2 (2%) | 2 (2%) | 4 (2%) |
| 3 | 1 (1%) | 4 (3%) | 5 (2%) |
| No of X-rays during hospital admission | | | |

Continued

**Table 4** Continued

| | Co-amoxiclav (N=133) | Placebo (N=132) | Total (N=265) |
|---|---|---|---|
| 0 | 132 (99%) | 130 (98%) | 262 (99%) |
| 1 | 1 (1%) | 2 (2%) | 3 (1%) |
| No of blood tests during hospital admission | | | |
| 0 | 133 (100%) | 130 (98%) | 263 (99%) |
| 1 | 0 (0%) | 2 (2%) | 2 (1%) |
| No of emergency department visits‡ | | | |
| 0 | 129 (97%) | 128 (97%) | 257 (97%) |
| 1 | 4 (3%) | 4 (3%) | 8 (3%) |

The summaries encompass the entire 28-day follow-up period, and may refer to more than one reconsultation episode.
The trial did not report any admissions to intensive care units.
*These summaries only include reconsultations that were not reported elsewhere in the form of hospital or emergency department admissions.
†Other interventions included blood and/or urine samples.
‡Participants who were reported to have been referred to the hospital team/ emergency department for acute admission during a reconsultation for clinical deterioration for the same illness episode for which the child was randomised, but for whom no hospital admission is recorded at this date, were classed as having visited emergency department. Hospital admission episodes for at least one night were collected during the medical notes review.
ARCHIE, Antibiotics for at Risk CHildren with InfluEnza-like illness.

Our findings, which present within-trial comparisons of costs and outcomes separately, are an important addition to the literature, due to the paucity of observational data and no previous clinical trials performed in this cohort of 'at-risk' children. Recruitment from both primary and other ambulatory care settings with wide geographical coverage and a very pragmatic definition of ILI ensured that our study population was highly representative of 'at-risk' children with ILI. High data availability rates for NHS costs as obtained via the medical notes review are also a strength of this study. Due to the low completion rate of the EQ-5D-Y, we did not perform a full cost-effectiveness analysis. Data on school/nursery attendance and child-care arrangements were also subject to low completion rates, and prevented a more robust assessment from a societal perspective.

The low response rates for the EQ-5D-Y and CARIFS data demonstrates the difficulties in assessing HRQoL in this patient population. Low questionnaire return rates are a common limitation of paediatric trials; for example, return rates at 14 days from two trials in respiratory tract infections were reported as 41%.[23] In our study, there was anecdotal evidence of parents and carers struggling to complete the questionnaires, especially the EQ-5D-Y questions on walking, usual activities and dressing, if the items were not appropriate to the expected capabilities of their children, either because of their age, especially for very young children, or their underlying condition or disability. This was reflected in the lower proxy completion rates of the EQ-5D-Y for children below the age of 2 years, compared with those aged 2 years and over, and the fact that the VAS scales of the EQ-5D-Y and the CARIFS had higher completion rates than the Likert-style questions of the questionnaires, as shown in the online supplemental material. Generally, there is a need for

validated questionnaires that can be completed easily in this patient population.

The EQ-5D-Y was expected to be completed by proxy, that is, by the children's parents or carers on their behalf, and/or by the children themselves, as appropriate. There appeared to be some overlap between the answers provided on the proxy and self-completed EQ-5D-Y questionnaires, particularly for very young children. It is possible that some parents or carers completed the questionnaires supposed to be filled in by the children. All EQ-5D-Y data were analysed as received, without exclusions.

The proxy-completed EQ-5D-Y values were not statistically significantly different between the two groups at any of the time points, although there was an improvement in HRQOL as indicated by the EQ-5D-Y scores over time (day 0–28) for both trial arms. There was a trend towards lower EQ-5D-Y values, indicating worse QoL, in the co-amoxiclav arm on days 4, 14 and 28. The CARIFS score in the co-amoxiclav arm at 7 days was significantly lower than the score in the placebo arm, indicating less severe symptoms. This might suggest that the CARIFS score is more sensitive to subtle disease specific changes in children's health than the EQ-5D-Y. Interestingly, the EQ-5D VAS also showed a statistically significant benefit of co-amoxiclav at 7 days, and the EQ-5D-Y values were higher (not statistically significant) in the co-amoxiclav arm at 7 days compared with the placebo arm, indicating higher HRQOL. Studies of common cold or non-specific respiratory tract infections in general paediatric populations suggest that symptoms in 50% of children had improved by day seven to eight, and for 80% by day 14.[24] It may, therefore, be that any differences between groups will be more apparent at a stage in the illness where more children are likely to still be symptomatic than later on,

**Table 5** Overview of costs per participant for reconsultations and hospital admissions related to index influenza-like illness episode*

| Items | Co-amoxiclav: Cost in £ (N=133) | | | | Placebo: Cost in £ (N=132) | | | | Difference† in cost in £ (95% CI) N=265 | |
|---|---|---|---|---|---|---|---|---|---|---|
| | Mean (SD) | Median | IQR | Range | Mean (SD) | Median | IQR | Range | Mean difference (95% CI) | P value |
| Intervention | 5 (1) | 4 | 4–6 | 4–6 | 0 (0) | 0 | 0–0 | 0–0 | 5 (95% CI 4 to 5) | <0.001 |
| Reconsultations | 16 (46) | 0 | 0–0 | 0–201 | 16 (54) | 0 | 0–0 | 0–402 | 1 (95% CI –9 to 10) | 0.910 |
| Antibiotics at reconsultation | 0 (1) | 0 | 0–0 | 0–10 | 0 (1) | 0 | 0–0 | 0–5 | 0 (95% CI 0 to 0) | 0.920 |
| Other drugs at reconsultation | 0 (2) | 0 | 0–0 | 0–13 | 1 (4) | 0 | 0–0 | 0–34 | 0 (95% CI –1 to 1) | 0.593 |
| X-rays at reconsultation | 7 (40) | 0 | 0–0 | 0–231 | 2 (20) | 0 | 0–0 | 0–231 | 5 (95% CI –2 to 13) | 0.178 |
| Other interventions at reconsultation | 0 (0) | 0 | 0–0 | 0–4 | 0 (0) | 0 | 0–0 | 0–0 | 0 (95% CI 0 to 0) | 0.210 |
| Hospital stay | 59 (421) | 0 | 0–0 | 0–4564 | 94 (484) | 0 | 0–0 | 0–4564 | –34 (95% CI –118 to 51) | 0.432 |
| Antibiotics during hospital stay | 0 (2) | 0 | 0–0 | 0–21 | 0 (0) | 0 | 0–0 | 0–3 | 0 (95% CI 0 to 1) | 0.247 |
| Other drugs during hospital stay | 0 (1) | 0 | 0–0 | 0–13 | 1 (3) | 0 | 0–0 | 0–34 | 0 (95% CI –1 to 0) | 0.283 |
| X-rays in hospital | 2 (20) | 0 | 0–0 | 0–231 | 4 (28) | 0 | 0–0 | 0–231 | –2 (95% CI –8 to 5) | 0.596 |
| Other interventions in hospital | 0 (0) | 0 | 0–0 | 0–0 | 0 (0) | 0 | 0–0 | 0–1 | 0 (95% CI 0 to 0) | 0.168 |
| Emergency department | 5 (1) | 0 | 0–0 | 0–160 | 5 (28) | 0 | 0–0 | 0–160 | 0 (95% CI –6 to 6) | 0.983 |
| Total cost | 94 (480) | 4 | 4–6 | 4–5258 | 122 (539) | 0 | 0–0 | 0–5177 | –25 (95% CI –113 to 62) | 0.566 |
| Total primary care costs | 8 (29) | 0 | 0–0 | 0–300 | 4 (12) | 0 | 0–0 | 0–63 | 3 (95% CI –2 to 8) | 0.215 |
| Total secondary care costs | 82 (481) | 0 | 0–0 | 0–5254 | 117 (540) | 0 | 0–0 | 0–5177 | –33 (95% CI –120 to 54) | 0.453 |

Reconsultations and related costs were attributed to primary care costs for children who had been recruited in a GP setting, and to secondary care costs for children who had been recruited in a paediatric ambulatory assessment unit. The primary and secondary care costs shown relate to the trial follow-up period, and do not include the trial intervention.

*Related to the illness episode for which the child was randomised into ARCHIE over the 28-day follow-up period.

†Differences have been adjusted for the minimisation factors age (used as continuous variable) and seasonal influenza vaccination status. Clustering by centre has been accounted for using the 'cluster' option in Stata's 'regress' command and robust standard errors were generated.

ARCHIE, Antibiotics for at Risk CHildren with InfluEnza-like Illness; GP, general practitioner.

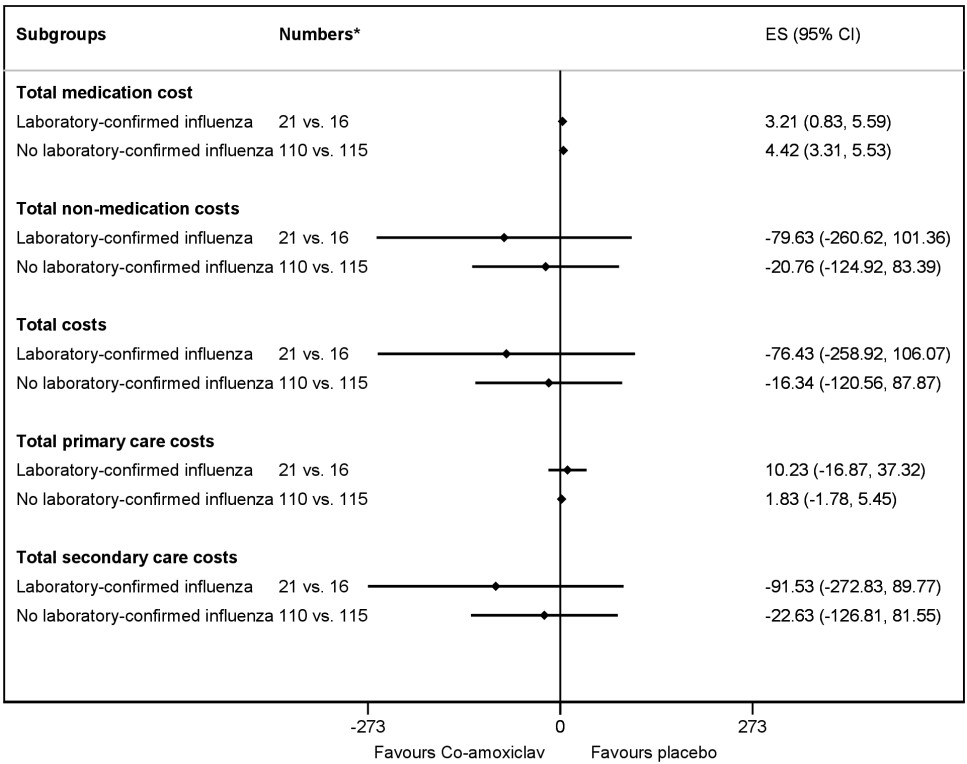

**Figure 1** Forest plot for differences in costs by laboratory-confirmed influenza (in £). *Subgroup effects were obtained by adding an interaction term between treatment allocation and laboratory-confirmed influenza and have been adjusted for the stratification factors age (used as continuous variable) and seasonal influenza vaccination status. Clustering by centre has been accounted for using the 'cluster' option in Stata's 'regress' command and robust standard errors were generated.

when the majority will have resolved due to the natural course of the illness. Both questionnaires were subject to high levels of missing data, and should be interpreted cautiously. The high rates of missing data and anecdotal evidence of parents' and carers' difficulties of completing the questionnaires demonstrate a need for additional validated tools to reliably measure QoL in this patient population. The EQ-5D-Y, EQ-5D-VAS and CARIFS were secondary endpoints of the ARCHIE trial, and the EQ-5D-Y was measured at several follow-up time points. As such, the limited number of statistically significant p values might be false positives due to multiple testing and should be interpreted carefully.

Overall, only 5% of children were admitted to hospital for reasons related to the illness episode for which the child was randomised into ARCHIE over the 28-day follow-up period. Admissions were often only for one or two nights. Three per cent of participants reported one emergency department visit each. Fewer than 25% of all randomised children had a reconsultation. This was lower than the anticipated 40% reconsultation rate based on published data available at the time the study was designed,[2] but still significantly higher than the 4% observed in a primary care cohort of children with acute respiratory tract infection who did not have known risk factors for complications from influenza or ILI.[25] As such, the use of primary and secondary care facilities during

the trial follow-up for reasons related to the initial illness episode for which the child was randomised into ARCHIE was generally low and the limited related hospital admissions were the main cost drivers for the average cost per randomised child. In the co-amoxiclav arm, costs related to hospital admissions and emergency department visits accounted for 70% of total costs; in the placebo arm, this was 85%.

Fourteen per cent of the study population had laboratory-confirmed influenza. We conducted a prespecified subgroup analysis in participants with laboratory-confirmed influenza because previous data suggesting a possible clinical benefit from early antibiotic use are reported in trials which recruited patients with confirmed influenza[26] or ILI presenting during an influenza epidemic,[27] when a high proportion of ILI cases are likely to be due to influenza. This subgroup analysis indicated that there was a trend that the early co-amoxiclav treatment may be more cost saving in participants with laboratory-confirmed influenza, although this subgroup does not have sufficient statistical power to draw definitive conclusions. Also, this trial assessed the benefit of treating at-risk children as early as possible after the onset of ILI. While there is a possibility that treatment may be more beneficial in those with laboratory-confirmed influenza, evaluating the nasal swabs takes time, and may make it impossible to treat these children early. Nasal swabs

were costed at £8 per child, and all children presenting with ILI would need to be tested. This outweighs the costs of co-amoxiclav, which was costed at £4.13 for children below 7 year of age, and at £5.79 children above 7 years of age. As such, there is no advantage in terms of costs to confirm influenza prior to commencing of treatment outside influenza pandemic periods. Based on the findings of the subgroup analysis, early co-amoxiclav treatment without prior nasal swab testing may be cost-effective during periods of high influenza activity, such as influenza pandemics. However, further trials conducted during such periods would be needed to confirm this.

In line with the study protocol, we aimed to extrapolate the results of the study to a national level. With regards to the ARCHIE trial, this extrapolation has a number of limitations. First, the difference in costs between the co-amoxiclav and placebo was not statistically significant. Therefore, it is unclear if extrapolation to a national level would result in a cost saving or increase in spending. Finally, influenza is an infectious disease, and rates of infections vary greatly between epidemics. Our current extrapolation uses GP consultations for ILI in 2018/2019. However, it is unclear how many children may be affected in upcoming influenza seasons. This is particularly important as the rates of influenza were low during the recruitment period of ARCHIE,[28–31] but more children may be affected in coming years. Our extrapolations were based on recent data, but may have to be revised in the light of SARS-CoV-2, and its impact on influenza and ILI in future influenza seasons.

Neither the findings of this economic evaluation nor the findings of our main trial[1] support early use of co-amoxiclav in 'at-risk' children who present with ILI in primary or ambulatory care.[1] With increasing concerns about growing antimicrobial resistance,[32 33] preserving antibiotic effectiveness by avoiding routine early co-amoxiclav in this group is therefore important. The findings of a longitudinal study nested within the ARCHIE trial will report data on long-term bacterial carriage and antibiotic resistance in the co-amoxiclav versus placebo arms in a separate paper. These findings may help inform estimations relating to the potential implications of antibiotic resistance on the results reported in this economic evaluation. However, incorporating the potential cost of antimicrobial resistance to the health system and society into economic evaluations is challenging.[32 34 35]

## CONCLUSIONS

Our findings did not show evidence that early treatment with co-amoxiclav in 'at-risk' children with ILI improves QoL, reduces healthcare use or leads to cost savings over 28 days. However, there was a suggestion that early co-amoxiclav treatment might be associated with decreased symptom severity at day 7, though confirmation from additional research would be important. Collection of proxy-reported data was challenging in this patient population. Suitable validated QoL instruments

are therefore needed to support more robust evaluations in this group.

**Acknowledgements** The research team acknowledges the support of the National Institute for Health Research Clinical Research Network (NIHR CRN). We would like to thank the ARCHIE Investigators for their support throughout the study, in particular: Malcolm G Semple PhD, Michael Moore FRCGP, Alastair D Hay FRCGP, Ushma Galal MSc, Tricia Taffe Carver BS, Rafael Perera-Salazar DPhil, Ly-Mee Yu DPhil, Jill Mollison PhD, Paul Little FRCGP, Andrew Farmer DM and Christopher Butler FMedSci. We thank Mandy Wan (Paediatric Research Pharmacist, NIHR CRN Children speciality) for her support and guidance regarding trial medication, from procurement to disposal. Staff in Oxford University's Primary Care Clinical Trials Unit were responsible for randomisation software design (David Judge), database development (Sadie Kelly), trial co-ordination (Sonya Beecher) and patient recruitment (Heather Rutter, Karen Madronal, Irene Noel, Luisa Saldana Ortega, Pippa Whitbread, Bernadette Mundy, Belinda I'Anson, Samantha Spires). We thank our Programme Steering Committee (Professor Willie Hamilton, Professor Judy Breuer, Professor Maureen Baker, Robert Newby), and Data and Safety Monitoring Committee (Professor Lorcan McGarvey, Mike Bradburn, Professor Surinder Birring, Professor Nick Francis), and our regional trial coordinators (Lyn Liddiard, Tammy Thomas, Clare McIntyre) for their valuable advice and support with delivering the trial. We thank Mildred Foster and Dan Richards-Doran for developing our study website and other study-related publicity materials. We also thank Dr Oliver van Hecke for his assistance with monitoring data reconsultation data extracted from medical records. Last but not least, we thank all the health care professionals, general practice and hospital staff, children and parents/guardians who participated in our study.

**Collaborators** ARCHIE (The early use of antibiotics for At-Risk CHildren with InfluEnza in primary care) investigators: Professor Anthony Harnden (Nuffield Department of Primary Care Health Sciences, University of Oxford, Oxford, UK), Dr Kay Wang (Nuffield Department of Primary Care Health Sciences, University of Oxford, Oxford, UK), Professor Malcolm G Semple (National Institute for Health Research (NIHR) Health Protection Research Unit in Emerging and Zoonotic Infections, University of Liverpool, Liverpool, UK), Dr Jane Wolstenholme (Health Economics Research Centre, University of Oxford, Oxford, UK), Professor Rafael Perera-Salazar (Nuffield Department of Primary Care Health Sciences, University of Oxford, Oxford, UK), Dr Ly-Mee Yu (Nuffield Department of Primary Care Health Sciences, University of Oxford, Oxford, UK), Professor Alastair D Hay (Centre for Academic Primary Care, School of Social and Community Medicine, University of Bristol, Bristol, UK), Professor Paul Little (Primary Care Population Sciences and Medical Education, University of Southampton, Southampton, UK), Professor Michael Moore (Primary Care Population Sciences and Medical Education, University of Southampton, Southampton, UK), Professor Chris Butler (Nuffield Department of Primary Care Health Sciences, University of Oxford, Oxford, UK), Tricia Taffe Carver (Nuffield Department of Primary Care Health Sciences, University of Oxford, Oxford, UK).

**Contributors** KW was chief investigator of the trial until going on maternity leave in September 2018. She reviewed clinical data on outcome events to guide collation and analysis of data for this study. ST was Trial Manager of the trial from 2017 until its conclusion. She managed the day to day data collection activities. JG was the data manager on the trial from January 2018 until its conclusion. AH was chief investigator on the trial from September 2018 until its conclusion. He reviewed clinical data on outcome events to guide collation and analysis of data for this study. JW was the senior health economist on the trial. She developed the economic evaluation plan, and led the economic analysis of the trial findings. IR conducted the analyses for this paper. IR and JW drafted the first version of the manuscript. All authors reviewed the manuscript, contributed comments and edits, and approved the final version.

**Funding** This project was funded by the National Institute for Health Research (NIHR) Programme Grants for Applied Research Programme (project reference RP-PG-1210-12012).

**Disclaimer** The views expressed are those of the authors and not necessarily those of the NIHR or the Department of Health and Social Care. The funder of the study had no role in design and conduct of the study; collection, management, analysis, and interpretation of the data; preparation, review, or approval of the manuscript; and decision to submit the manuscript for publication.

**Competing interests** KW held a National Institute for Health Research (NIHR) Academic Clinical Lectureship during the conduct of the study and currently holds an NIHR Postdoctoral Fellowship.

**Patient and public involvement** Patients and/or the public were involved in the design, or conduct, or reporting, or dissemination plans of this research. Refer to the Methods section for further details.

**Patient consent for publication** Not applicable.

**Ethics approval** The ARCHIE trial has been reviewed and approved by the North West-Liverpool East Research Ethics Committee (13/NW/0621), Health Research Authority and Medicines and Healthcare Products Regulatory Agency. Full written informed consent was obtained from a parent or guardian for each study participant. Children were also invited to give written assent if appropriate. Participants gave informed consent to participate in the study before taking part.

**Provenance and peer review** Not commissioned; externally peer reviewed.

**Data availability statement** Data are available on reasonable request. Data are available on reasonable request. Research data requests should be submitted to information.guardian@phc.ox.ac.uk for consideration.

**ORCID iDs**
Ines Rombach http://orcid.org/0000-0003-3464-3867
Kay Wang http://orcid.org/0000-0002-7195-1730
Sharon Tonner http://orcid.org/0000-0002-7775-9926
Jenna Grabey http://orcid.org/0000-0001-7498-1276
Anthony Harnden http://orcid.org/0000-0003-0013-9611
Jane Wolstenholme http://orcid.org/0000-0001-7493-1850

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
