## [Reviewer comments · BMJ Open]

ARTICLE DETAILS

TITLE (PROVISIONAL)	Quality of life, healthcare use and costs in 'at risk' children after early antibiotic treatment vs. placebo for influenza-like illness: Within-trial descriptive economic analyses of the ARCHIE randomised controlled trial
AUTHORS	Rombach, Ines; Wang, Kay; Tonner, Sharon; Grabey, Jenna; Harnden, Anthony; Wolstenholme, Jane

VERSION 1 – REVIEW

REVIEWER	Cooper, Celia Womens and Childrens Hospital (WCH)
REVIEW RETURNED	29-Mar-2021

GENERAL COMMENTS	This is an interesting idea to study. Probably good that your data don't support wholesale prescribing of co-amoxiclav to all children presenting with ILI as this would not support antimicrobial stewardship initiatives which I thought had been a major focus of the UK and EU. In fact, I wonder if the need to preserve antibiotic effectiveness by reducing unnecessary use given the lack of evidence of benefit in your study could be included in your discussion? I think that you need to be careful making too much out of a single, barely significant finding of improvement (Day 7 CARIFS) when there was no other evidence of improvement in any other scores. I don't know what the relatedness is between the measures but since you are essentially measuring similar things using a variety of tools shouldn't use some sort of correction (e.g. Bonferroni correction or similar) rather than an ordinary p-value? I am not a statistician but this struck me as an inappropriate use of statistics since it seems that a single measure of improvement might be more likely to occur by chance if you are using multiple related measures of the same outcome. Thus I'm not sure that you can say "However, early co-amoxiclav treatment was associated with decreased symptom severity at day 7" unless you also add a qualifying statement like " though this was not reflected in other similar measures". I also don't think that you can say that that the estimated national cost savings of using co-amoxiclav treatment is 144,675 pounds. The confidence interval shows that the impact could be anything between a saving of 358,794 pounds to an increased cost of 653,931 pounds. These are the numbers that decision makers should be focusing on, not a putative saving that may never materialise based on your data. I think you made some interesting points. You certainly provided support for the need to develop new measurements for cost effectiveness in paediatric interventions as there are clearly major difficulties with the application of current ones.
--

REVIEWER	Robertson, Iain University of Tasmania, College of Health and Medicine
REVIEW RETURNED	07-Jul-2021

GENERAL COMMENTS	Reviewer comments on manuscript BMJ_Open_049343. "Quality of life, healthcare use and costs in 'at risk' children after early antibiotic treatment vs. placebo for influenza-like illness: Within-trial economic analyses of the ARCHIE randomised controlled trial." Reviewer: Iain K Robertson, Adjunct Senior Researcher, School of Health Sciences, University of Tasmania, Launceston, Australia Overall comments: The subject of the principal randomised controlled trial, whether and how antibiotics should be used to treat a probable viral infection in a group of patients potentially more vulnerable to the effects of that illness, is a question worthy of resolving. As with all treatments, the circumstances of the use of the treatment needs to be judged with an understanding of the benefits and costs in a form that allows the relative value of this treatment to be judged against any other alternative use of the resources applied. The central core of the clinical dilemma is the development of antibiotic resistance if antibiotics are overused, particularly in circumstances like viral illness where little benefit can be expected. This might render antibiotics less effective to future patients who might gain greater benefit from their use, instead of their use where little benefit it gained. Yet antibiotic resistance is not mentioned once in the manuscript. An economic evaluation is a conceptual examination of what might happen in the future. The particular circumstances of a randomised controlled trial may be used as an example of what might happen, even though it has happened in the past, and we are only interested in what will happen in the future. Any judgement of value will require bringing together information from a variety of sources with the aim of being comprehensive about the influences impacting those judgements. Very little of the consideration of what was measured, and what more needs to be inferred from other studies, is included in this manuscript. What are the benefits or otherwise of the treatment being considered, what are the resources used, who receives the benefits, and who pays the costs? In its current form, the manuscript is an inadequate consideration of the issues that clinicians and health service managers need to consider when making their decisions. The manuscript is not presented in compliance with the CHEERS reporting guidelines. I have used the structure of those guidelines as a template for my comments. My comments are likely not comprehensive, but I hope that the structure will indicate where further manuscript development may be needed, at the discretion of the authors. CHEERS checklist—Items to include when reporting economic evaluations of health interventions: BMJ 2013; 346: f1049 Section/item Item No Recommendation
---

	Review comments Title and abstract Title 1 Identify the study as an economic evaluation or use more specific terms such as “cost-effectiveness analysis”, and describe the interventions compared. The title is not precise about what economic analysis is being performed. Abstract 2 Provide a structured summary of objectives, perspective, setting, methods (including study design and inputs), results (including base case and uncertainty analyses), and conclusions. The abstract is reflective of the difficulties that the authors had in conducting the randomised controlled trial, and the limited conclusions again reflect the limitations of the study design and the analysis they felt able to conduct. However, there was also a limitation of conceptual analysis of the clinical situation, and the issues involved, that undermines the value of what they were actually able to say about the RCT results (see below for details of concerns). Introduction Background and objectives 3 Provide an explicit statement of the broader context for the study. There are four main contexts surrounding the target treatment in the target population: Does the treatment improve the adverse event rates (as measured by the re-consultation rate) or symptom profile (the utility measures chosen); Does the treatment (the specific antibiotic) have an adverse effect on the nature of the biofilms carried by the participants being treated (are they at risk of developing separate microbial effects from the derangement of resident microbe populations by the antibiotics); What is the contribution of the use of antibiotics in this group of health service users on the development of antibiotic resistance? The psychology of decision-making is likely to be complex: The antibiotic may be a way that does not require extended counselling of addressing parental concerns; There is always the possibility of an uncommon nasty bacterial infection (e.g. meningitis) mimicking the viral illness, and
--	---

	explaining and organising a specific targeted response to this may cause even greater parental anxiety; The doctor may feel that the patient before them is their primary responsibility, and that other more general risks are for a later time. The trial appears to have been designed to assess only the first of these questions. For an economic evaluation, it is important to identify significant long-term effects on the individuals being treated, and any externalities experienced by the general population. It is my understanding that there has been a longstanding concern about the use of antibiotics to treat viral infections, and as prophylaxis for secondary bacterial infections in viral diseases, due to the potential and actual development of antibiotic resistance. This would seem to be a highly relevant issue for an economic evaluation of the treatment being studied: who gains the benefits, and who pays the price. It may not be possible to measure the economic impact of the risk of development of antibiotic resistance of any particular instance of antibiotic use. However, the failure to make any comment on the issue suggests that the authors judge that this issue has no impact and no cost implications. I doubt that this is actually the position of the authors, so it would be helpful if they address this issue to the best of their ability. The estimation of the effects of antibiotic resistance is likely not to be subject to linear effects, but more likely to arise from chance (Poisson-distribution) occurrences that expand exponentially at unpredictable rates. The mathematics of health economic analysis usually use linear-type equations to carry their principal meanings. These uncertainties make judgements difficult, but sensitivity analysis may be helpful in reducing the chaos: a series of what-ifs may be tested in simulation analysis. Antibiotic resistance is likely to be only very weakly relatable to any particular instance of antibiotic use. However, it is something like gravity: it is a very weak force taken by itself, but when aggregated, it can produce some of the most powerful forces in the universe; the usefulness of antibiotics can be frittered away by use in circumstances where their benefits are minimal. I would expect the authors to have a position on this question. Economics is a social science that studies the behaviour of people in circumstances where judgement about risks/benefits against resource use are made, and who are the actors involved in each of these components of the balance. The subject of this paper would seem highly appropriate to be addressed by such considerations, yet there is little attempt to engage with these issues. Present the study question and its relevance for health policy or practice decisions. As the manuscript stands, this absence of any consideration of antibiotic resistance means that it has no value for health policy or practice decisions. Methods
--	--

	Target population and subgroups 4 Describe characteristics of the base case population and subgroups analysed, including why they were chosen. There is inadequate description of the nature of the trial participants: a table of selection criteria; and numbers of actual case diagnoses (or whatever other relevant descriptions are available) would be helpful for the readers to get a sense of who the treatment is intended. It is not adequate to refer to another paper for this important issue, since many readers will not have the time to be chasing other papers. This might be included in the supplementary material, with brief reference in the main text. I am left with the question as a reader of whether I am missing particular reasons why this group of children are receiving antibiotics whilst most children might not. No explanation was given for the size of the sample of the main trial. Presumably the calculation was made on the basis of an assumption of an expected rate of re-consultations. My calculations suggest that with an expectation of 40% re-consultations, a minimum clinically relevant relative difference being sought was 40%, which is quite large. There would be only a 40% power to detect a 25% reduction, which would still be a valuable effect. The actual re-consultation rate was about 24%, and this reduced the power of the trial to 56% at the original 40% minimum effect size, or 23% at a more realistic 25% effect size. Thus, the trial seems underpowered to detect the primary outcome of the trial. An economic evaluation involves variability in both the benefits and costs of a treatment. Thus, the power of the trial will inevitably be even lower than the plan for the detection of just the benefits. No adequate comments are made concerning these issues. It is accepted that no helpful further analysis can make up for the inadequate numbers, but comments on this issue may guide the readers in judging the context of the results. Setting and location 5 State relevant aspects of the system(s) in which the decision(s) need(s) to be made. We are referred to the protocol for the study detail, but no adequate explanation is given for how the decisions were being made. Study perspective 6 Describe the perspective of the study and relate this to the costs being evaluated. Again, this issue is not handled adequately, since this involves a discussion of value judgements which have not been rehearsed in this manuscript.
--	--

	Comparators 7 Describe the interventions or strategies being compared and state why they were chosen. This seems minimalist, due to an apparent reliance on the protocol and the primary outcome paper (it is unclear whether the latter has been published as of 28th June 2021, accepted for publication or other publication status, which eventually will need to be corrected if this manuscript is not to become an orphan publication). Time horizon 8 State the time horizon(s) over which costs and consequences are being evaluated and say why appropriate. This is not adequately addressed, due to the deficiencies of what is written in the “Background and Objectives” section Discount rate 9 Report the choice of discount rate(s) used for costs and outcomes and say why appropriate. Comment is made, which may be adequate Choice of health outcomes 10 Describe what outcomes were used as the measure(s) of benefit in the evaluation and their relevance for the type of analysis performed. This is minimally addressed, but may be improved by addressing deficiencies in other sections of the manuscript. Measurement of effectiveness 11a Single study-based estimates: Describe fully the design features of the single effectiveness study and why the single study was a sufficient source of clinical effectiveness data. This is minimally addressed, but may be improved by addressing deficiencies in other sections of the manuscript. 11b Synthesis-based estimates: Describe fully the methods used for identification of included studies and synthesis of clinical effectiveness data.
--	--

	This is minimally addressed. The absence of discussion of the wider context of the use of the antibiotics in this clinical context prevents a full appreciation of any of the comments that were made. Measurement and valuation of preference- based outcomes 12 If applicable, describe the population and methods used to elicit preferences for outcomes. This does not seem to have been considered to be part of this study. Estimating resources and costs 13a Single study-based economic evaluation: Describe approaches used to estimate resource use associated with the alternative interventions. Describe primary or secondary research methods for valuing each resource item in terms of its unit cost. Describe any adjustments made to approximate to opportunity costs. This is not addressed in this manuscript. 13b Model-based economic evaluation: Describe approaches and data sources used to estimate resource use associated with model health states. Describe primary or secondary research methods for valuing each resource item in terms of its unit cost. Describe any adjustments made to approximate to opportunity costs. This approach was not chosen. Currency, price date, and conversion 14 Report the dates of the estimated resource quantities and unit costs. Describe methods for adjusting estimated unit costs to the year of reported costs if necessary. Describe methods for converting costs into a common currency base and the exchange rate. Minimally addressed, although this is not a major deficiency of the manuscript Choice of model 15 Describe and give reasons for the specific type of decision-analytical model used. Providing a figure to show model structure is strongly recommended. Not addressed
--	---

	Assumptions 16 Describe all structural or other assumptions underpinning the decision-analytical model. Not addressed Analytical methods 17 Describe all analytical methods supporting the evaluation. This could include methods for dealing with skewed, missing, or censored data; extrapolation methods; methods for pooling data; approaches to validate or make adjustments (such as half cycle corrections) to a model; and methods for handling population heterogeneity and uncertainty. It appears that the statistical analyses conducted used multiple linear regression with clustering on GP practice. This may be an adequate approach, assuming that any distributional and assumption non-compliance issues were explored during the analytic process. The data being examined are mostly rank-ordered in nature, which is conventionally aggregated into scores that are analysed with such linear regression methods. If any doubts about the appropriateness of this usage, it can be checked using a rank-ordered equivalent, such as ordered logistic regression (ologit): if no major difference in interpretation arises from the use of ologit, the results of linear regression analysis are easier for a reader to assimilate. Thus, what is described is in line with my practice in these circumstances. Were the covariate adjustments conducted using raw values, or standardised normal transformations (z-scores)? The way that covariate regression models work is that the constant is adjusted to the mean value of the outcome when all the covariates are zero. This would mean that raw age adjustment would tell you what the judgement of the child when treated at the time of their birth. The standardised normal transformation ((Value minus mean)/standard deviation) has, by construction, a mean of zero and a standard deviation of 1. If all covariates, excluding the treatment group in an RCT for example, were presented as z-scores, the regression constant would be in the middle of the data distribution (rather than outside the distribution), and all covariates would be corrected for differing scales, having a standard deviation of 1 standard deviation. [I include this comment only for clarification. Interpretation of numbers is a very fiddly thing, of which details many readers may be unaware.] The comment about clustering on general practices raises the question about whether random effects might have been operating, which might have been addressed by random effects modelling ("mixed" Stata syntax). The conduct and interpretation of the analysis would not be greatly altered by inclusion of this if random effects variable might have been operating and measured. There was a large amount of missing data. No explanation of how this was handled is made in the analytic methods section, despite the apparent claim that the analysis was performed on an
--	---

	intention-to-treat basis. This should be clarified, and the analysis revised if necessary. The sensitivity analysis, looking at the differential effects in influenza-confirmed and -unconfirmed cases seems appropriate, although the problem of inadequate sample size is acknowledged. It should be noted that no attempt was reported in the manuscript to estimate cost-utility parameters: one method is to plot utility benefits (marginal utility on x-axis) against costs (marginal costs on y-axis), with levels of variability defined using bootstrap simulation from the trial results. This would make clear the degree of uncertainty that surrounds any judgements being made from a single trial. Results Study parameters 18 Report the values, ranges, references, and, if used, probability distributions for all parameters. Report reasons or sources for distributions used to represent uncertainty where appropriate. Providing a table to show the input values is strongly recommended. It would seem appropriate to have a brief statement about the impact of the randomised treatments on the primary outcome of the trial. This is not included early in the results section. My suspicion is that this arises from the queuing of papers from the trial presented for publication. Has the main paper been published or accepted for publication? This might explain this deficit, not wanting to pre-empt publication of the main results. Otherwise, there is no reason not to make a brief comment. Incremental costs and outcomes 19 For each intervention, report mean values for the main categories of estimated costs and outcomes of interest, as well as mean differences between the comparator groups. If applicable, report incremental cost-effectiveness ratios. This is presented for the utility/QoL/symptom score measures in a simple form. Characterising uncertainty 20a Single study-based economic evaluation: Describe the effects of sampling uncertainty for the estimated incremental cost and incremental effectiveness parameters, together with the impact of methodological assumptions (such as discount rate, study perspective). The effects of sampling uncertainty on estimated incremental costs were not adequately handled. The costs are predominantly borne by the health system. As such, total costs, and mean cost per patient, are the relevant quantities to be reported. The distribution of these will be skewed, and so estimation of asymmetrical confidence intervals would be appropriate. This can be achieved
--	--

by bootstrap multivariate linear regression with estimation of percentile 95% confidence intervals: (“bootstrap, bca:” syntax is placed before the regression model syntax, followed by “estat bootstrap, percentile” to get a table with the percentile 95% CIs, after checking whether this syntax matches your previous syntax). The study perspective used represents an inadequate handling of this particular clinical situation, as commented above.
20b

Model-based economic evaluation: Describe the effects on the results of uncertainty for all input parameters, and uncertainty related to the structure of the model and assumptions.

Characterising heterogeneity

21

If applicable, report differences in costs, outcomes, or cost-effectiveness that can be explained by variations between subgroups of patients with different baseline characteristics or other observed variability in effects that are not reducible by more information.

It is unclear why the choice of influenza-confirmed is reported in the sensitivity analysis. Other subgroups, such as specific disease-related groups might also have been considered, although it is likely that the cases numbers for these subgroups would be too small for confidence in the results. The absence of information about these clinical subgroups in the manuscript prevents a definitive judgement on this question.

Discussion

Study findings, limitations, generalisability, and current knowledge

22

Summarise key study findings and describe how they support the conclusions reached. Discuss limitations and the generalisability of the findings and how the findings fit with current knowledge.

The previous comments suggest that there has been inadequate discussion of the results of the study. Any conclusions will be of limited applicability and generalisability.

It is possibly/likely, given the low numbers of cases and inadequate economic perspective contained in the data collected as part of the study, that the main value of a manuscript arising from the trial, would be a discussion of how the question of the utility and cost-utility of the use of antibiotics (and this specific antibiotic) in influenza-like illnesses in “at-risk” children (however these are defined) might be evaluated in order to assist the health system as a whole to judge when and how those antibiotics should be used. What is provided in the manuscript is, up to this time, inadequate for forming those judgements. The authors might usefully consider how such ideas might be rehearsed in a revised manuscript.

	Other Source of funding 23 Describe how the study was funded and the role of the funder in the identification, design, conduct, and reporting of the analysis. Describe other non-monetary sources of support. Conflicts of interest 24 Describe any potential for conflict of interest of study contributors in accordance with journal policy. In the absence of a journal policy, we recommend authors comply with International Committee of Medical Journal Editors recommendations. For consistency, the CHEERS statement checklist format is based on the format of the CONSORT statement checklist
--	--

VERSION 1 – AUTHOR RESPONSE

Reviewer: 1

Dr. Celia Cooper, Womens and Childrens Hospital (WCH) Comments to the Author:

This is an interesting idea to study.

Probably good that your data don't support wholesale prescribing of co-amoxiclav to all children presenting with ILI as this would not support antimicrobial stewardship initiatives which I thought had been a major focus of the UK and EU.

In fact, I wonder if the need to preserve antibiotic effectiveness by reducing unnecessary use given the lack of evidence of benefit in your study could be included in your discussion?

Authors' response:

We have now included the following into our discussion session:

“Neither the findings of this economic evaluation nor the findings of our main trial (Wang K, Semple MG, Moore M, et al. The early use of Antibiotics for at Risk CHildren with Influenza-like illness (ARCHIE): a double-blind randomised placebo-controlled trial. Eur Respir J 2021) support early use of co-amoxiclav in ‘at risk’ children who present with ILI in primary or ambulatory care.

With increasing concerns about growing antimicrobial resistance, preserving antibiotic effectiveness by avoiding routine early co-amoxiclav in this group is therefore important. ”

I think that you need to be careful making too much out of a single, barely significant finding of improvement (Day 7 CARIFS) when there was no other evidence of improvement in any other scores. I don't know what the relatedness is between the measures but since you are essentially measuring similar things using a variety of tools shouldn't use some sort of correction (e.g. Bonferroni correction

or similar) rather than an ordinary p-value? I am not a statistician but this struck me as an inappropriate use of statistics since it seems that a single measure of improvement might be more likely to occur by chance if you are using multiple related measures of the same outcome. Thus I'm not sure that you can say "However, early co-amoxiclav treatment was associated with decreased symptom severity at day 7" unless you also add a qualifying statement like " though this was not reflected in other similar measures".

Authors' response: This is a very valid point, and we have made these limitations clearer in the abstract, discussion and conclusion sections.

Addition to the discussion:

"Also, the EQ-5D-Y, EQ-5D-VAS and CARIFS were secondary endpoints of the ARCHIE trial, and the EQ-5D was measured at several follow-up timepoints. As such, the limited number of statistically significant p-values might be false positives due to multiple testing and should be interpreted carefully."

Revision of the statement on potential decrease in symptom severity in the conclusion:

"However, there was a suggestion that early co-amoxiclav treatment might be associated with decreased symptom severity at day 7, though confirmation from further research would be important."

Abstract: "Our findings did not show evidence that early co-amoxiclav treatment improves quality of life or reduces healthcare use and costs in 'at risk' children with ILI, but may reduce symptom severity though confirmation from further research would be important."

I also don't think that you can say that that the estimated national cost savings of using co-amoxiclav treatment is 144,675 pounds. The confidence interval shows that the impact could be anything between a saving of 358,794 pounds to an increased cost of 653,931 pounds. These are the numbers that decision makers should be focusing on, not a putative saving that may never materialise based on your data.

Authors' response: We feel that our text on this is appropriate. We present the point estimate with corresponding confidence interval in the last paragraph of the results section : "...the potential cost saving of early co-amoxiclav treatment compared with not providing early antibiotics is estimated as £-144,675 (95% CI £-358,794 to £653,931)."

In the discussion, we state that "it is unclear if extrapolation to a national level would result in a cost-saving or increase in spending"

Our conclusions state that "Our findings did not show evidence that early treatment with co-amoxiclav in 'at risk' children with influenza-like illness improves quality of life, reduces healthcare use or leads to cost savings over 28 days. "

I think you made some interesting points. You certainly provided support for the need to develop new measurements for cost effectiveness in paediatric interventions as there are clearly major difficulties with the application of current ones.

Reviewer: Iain K Robertson, Adjunct Senior Researcher, School of Health Sciences, University of Tasmania, Launceston, Australia

Overall comments:

1. The subject of the principal randomised controlled trial, whether and how antibiotics should be used to treat a probable viral infection in a group of patients potentially more vulnerable to the effects of that illness, is a question worthy of resolving. As with all

treatments, the circumstances of the use of the treatment needs to be judged with an understanding of the benefits and costs in a form that allows the relative value of this treatment to be judged against any other alternative use of the resources applied.

Authors: no response required

2. The central core of the clinical dilemma is the development of antibiotic resistance if antibiotics are overused, particularly in circumstances like viral illness where little benefit can be expected. This might render antibiotics less effective to future patients who might gain greater benefit from their use, instead of their use where little benefit it gained. Yet antibiotic resistance is not mentioned once in the manuscript.

Authors' response: Thank you for this insightful suggestion.

Incorporating the potential cost of antimicrobial resistance to health system into economic evaluations is challenging, as documented by the following authors:

Coast J, Smith R, Karcher AM, Wilton P, Millar M. Superbugs II: how should economic evaluation be conducted for interventions which aim to contain antimicrobial resistance? *Health Econ* 2002; 11(7): 637-47.

Coast J, Smith RD, Millar MR. Superbugs: should antimicrobial resistance be included as a cost in economic evaluation? *Health Econ* 1996; 5(3): 217-26.

Jit M, Ng DHL, Luangasanatip N, et al. Quantifying the economic cost of antibiotic resistance and the impact of related interventions: rapid methodological review, conceptual framework and recommendations for future studies. *BMC Med* 2020; 18(1): 38.

We have added the following considerations to the end of the discussion section:

“With increasing concerns about growing antimicrobial resistance, preserving antibiotic effectiveness by avoiding routine early co-amoxiclav in this group is therefore important. The findings of a longitudinal study nested within the ARCHIE trial will report data on long-term bacterial carriage and antibiotic resistance in the co-amoxiclav versus placebo arms in a separate paper. These findings may help inform estimations relating to the potential implications of antibiotic resistance on the results reported in this economic evaluation..”

3. An economic evaluation is a conceptual examination of what might happen in the future. The particular circumstances of a randomised controlled trial may be used as an example of what might happen, even though it has happened in the past, and we are only interested in what will happen in the future. Any judgement of value will require bringing together information from a variety of sources with the aim of being comprehensive about the influences impacting those judgements. Very little of the consideration of what was measured, and what more needs to be inferred from other studies, is included in this manuscript. What are the benefits or otherwise of the treatment being considered, what are the resources used, who receives the benefits, and who pays the costs? In its current form, the manuscript is an inadequate consideration of the issues that clinicians and health service managers need to consider when making their decisions.

Authors' response:

Our analysis was a within-trial analysis from the NHS perspective, based on the data collected in the trial. We hope that we have made this clear in the manuscript. We have therefore focussed on the effect of the intervention on children's quality of life, symptoms and NHS costs. In additional analyses, we have reported some impacts on family units (i.e. days of school missed, days of work missed, change of usual activity and requirements for outside carers), though these analyses are

limited by high rates of missing data, in line with the other self-reported data for this trial. We have also now included further considerations on the effects of antimicrobial resistance.

Section/item	Item No	Recommendation	Reviewer's comment	Author response
Title and abstract				
Title	1	Identify the study as an economic evaluation or use more specific terms such as "cost-effectiveness analysis", and describe the interventions compared.	The title is not precise about what economic analysis is being performed.	We are not performing a formal economic analyses and have amended our manuscript title accordingly. We describe our results now as a "descriptive economic analyses" to indicate that we are neither doing a cost-effectiveness, cost-benefit, cost-utility analysis etc.
Abstract	2	Provide a structured summary of objectives, perspective, setting, methods (including study design and inputs), results (including base case and uncertainty analyses), and conclusions.	The abstract is reflective of the difficulties that the authors had in conducting the randomised controlled trial, and the limited conclusions again reflect the limitations of the study design and the analysis they felt able to conduct. However, there was also a limitation of conceptual analysis of the clinical situation, and the issues involved, that undermines the value of	We have now added consideration of the study results in the context of antimicrobial resistance to the discussion section and in the abstract. We have added additional considerations of the study results in the light of antimicrobial resistance to the discussion (see responses above).

			what they were actually able to say about the RCT results (see below for details of concerns).	
Introduction				
Background and objectives	3	Provide an explicit statement of the broader context for the study. Present the study question and its relevance for health policy or practice decisions.	There are four main contexts surrounding the target treatment in the target population: 1. Does the treatment improve the adverse event rates (as measured by the re-consultation rate) or symptom profile (the utility measures chosen); 2. Does the treatment (the specific antibiotic) have an adverse effect on the nature of the biofilms carried by the participants being treated (are they at risk of developing separate microbial effects from the derangement of resident microbe populations by the antibiotics);	The reviewer makes interesting observations. We have now added considerations of the study results in the context of antimicrobial resistance to the discussion section (as mentioned above). Given the uncertainty around the current study results, we feel that additional sensitivity analyses will not be adding important insight to this question and believe that the added discussion is sufficient for the context of this study.

			3. What is the contribution of the use of antibiotics in this group of health service users on the development of antibiotic resistance? 4. The psychology of decision-making is likely to be complex: a. The antibiotic may be a way that does not require extended counselling of addressing parental concerns; b. There is always the possibility of an uncommon nasty bacterial infection (e.g. meningitis) mimicking the viral illness, and explaining and organising a specific targeted response to this may cause even greater parental anxiety; c. The doctor may feel that the patient before them is their	
--	--	--	--	--

			primary responsibility, and that other more general risks are for a later time. The trial appears to have been designed to assess only the first of these questions. For an economic evaluation, it is important to identify significant long-term effects on the individuals being treated, and any externalities experienced by the general population. It is my understanding that there has been a longstanding concern about the use of antibiotics to treat viral infections, and as prophylaxis for secondary bacterial infections in viral diseases, due to the potential and actual development of antibiotic resistance. This would seem to be a	
--	--	--	--	--

			highly relevant issue for an economic evaluation of the treatment being studied: who gains the benefits, and who pays the price. It may not be possible to measure the economic impact of the risk of development of antibiotic resistance of any particular instance of antibiotic use. However, the failure to make any comment on the issue suggests that the authors judge that this issue has no impact and no cost implications. I doubt that this is actually the position of the authors, so it would be helpful if they address this issue to the best of their ability. The estimation of the effects of antibiotic resistance is likely not to be subject to linear effects, but more likely to arise from	
--	--	--	--	--

			chance (Poisson-distribution) occurrences that expand exponentially at unpredictable rates. The mathematics of health economic analysis usually use linear-type equations to carry their principal meanings. These uncertainties make judgements difficult, but sensitivity analysis may be helpful in reducing the chaos: a series of what-ifs may be tested in simulation analysis. Antibiotic resistance is likely to be only very weakly relatable to any particular instance of antibiotic use. However, it is something like gravity: it is a very weak force taken by itself, but when aggregated, it can produce some of the most powerful forces in	
--	--	--	---	--

			the universe; the usefulness of antibiotics can be frittered away by use in circumstances where their benefits are minimal. I would expect the authors to have a position on this question. Economics is a social science that studies the behaviour of people in circumstances where judgement about risks/benefits against resource use are made, and who are the actors involved in each of these components of the balance. The subject of this paper would seem highly appropriate to be addressed by such considerations, yet there is little attempt to engage with these issues. As the manuscript stands, this absence of any consideration	
--	--	--	---	--

			of antibiotic resistance means that it has no value for health policy or practice decisions.	
Methods				
Target population and subgroups	4	Describe characteristics of the base case population and subgroups analysed, including why they were chosen.	1. There is inadequate description of the nature of the trial participants: a table of selection criteria; and numbers of actual case diagnoses (or whatever other relevant descriptions are available) would be helpful for the readers to get a sense of who the treatment is intended. It is not adequate to refer to another paper for this important issue, since many readers will not have the time to be chasing other papers. This might be included in the supplementary material, with brief reference in the maintext. I am left with the question as a	We agree, and have added the requested information to the manuscript, with reference to the main clinical paper. 1. We have added additional detail in the methods section of the manuscript on the inclusion/ exclusion criteria. we have also added a table of key baseline criteria to the manuscript (new Table 1). 2. We clarified the target sample size and corresponding calculations for the trial, and emphasised that the target sample size was not reached due to slower than anticipated recruitment during the available recruitment seasons.

			reader of whether I am missing particular reasons why this group of children are receiving antibiotics whilst most children might not. 2. No explanation was given for the size of the sample of the main trial. Presumably the calculation was made on the basis of an assumption of an expected rate of re-consultations. My calculations suggest that with an expectation of 40% reconsultations , a minimum clinically relevant relative difference being sought was 40%, which is quite large. There would be only a 40% power to detect a 25% reduction, which would still be a valuable effect. The actual re-consultation rate was	
--	--	--	--	--

			about 24%, and this reduced the power of the trial to 56% at the original 40% minimum effect size, or 23% at a more realistic 25% effect size. Thus, the trial seems underpowered to detect the primary outcome of the trial. An economic evaluation involves variability in both the benefits and costs of a treatment. Thus, the power of the trial will inevitably be even lower than the plan for the detection of just the benefits. No adequate comments are made concerning these issues. It is accepted that no helpful further analysis can make up for the inadequate numbers, but comments on this issue may guide the readers in	
--	--	--	---	--

			judging the context of the results.	
Setting and location	5	State relevant aspects of the system(s) in which the decision(s) need(s) to be made.	We are referred to the protocol for the study detail, but no adequate explanation is given for how the decisions were being made.	We have clarified that participants were recruited from general practices, walk-in centres and hospitals in the UK with a view to making healthcare decisions in the NHS and other healthcare settings.
Study perspective	6	Describe the perspective of the study and relate this to the costs being evaluated.	Again, this issue is not handled adequately, since this involves a discussion of value judgements which have not been rehearsed in this manuscript.	We have clarified that costs pertain to the UK NHS perspective.
Comparators	7	Describe the interventions or strategies being compared and state why they were chosen.	This seems minimalist, due to an apparent reliance on the protocol and the primary outcome paper (it is unclear whether the latter has been published as of 28th June 2021, accepted for publication or other publication status, which eventually will need to be corrected if this manuscript is not to	The primary results have now been published (https://erj.ersjournals.com/content/early/2021/03/02/13993003.02819-2020), and the reference to these results has been included in the updated manuscript We have reiterated the comparators and rationale in the study manuscript. Added information: "Children were randomised 1:1 to either co-amoxiclav 400/57 (amoxicillin 400 mg as trihydrate and clavulanic acid 57 mg as potassium salt/5 mL when reconstituted with water) or a matching placebo, which will be taken orally twice daily for 5 days. Co-amoxiclav was used due to its susceptibility in treating lower respiratory tract bacterial isolates associated with influenza in primary care {Blackburn, 2011 #202}, and is stockpiled by the UK government for use in influenza pandemics;, and a matching placebo was chosen to obtain unbiased results for treatment effects."

			become an orphan publication).	
Time horizon	8	State the time horizon(s) over which costs and consequences are being evaluated and say why appropriate.	This is not adequately addressed, due to the deficiencies of what is written in the "Background and Objectives" section	We clarify in the methods section that data are collected over a 28 day period. We have also reiterated at the beginning of the methods section that the time horizon of the study is 28 days.
Discount rate	9	Report the choice of discount rate(s) used for costs and outcomes and say why appropriate.	Comment is made, which may be adequate	No changes requested
Choice of health outcomes	10	Describe what outcomes were used as the measure(s) of benefit in the evaluation and their relevance for the type of analysis performed	This is minimally addressed, but may be improved by addressing deficiencies in other sections of the manuscript.	We have described the outcomes included in this manuscript in the section on Quality of life, CARIFs, health care use, and unit costs.
Measurement of effectiveness	11a	Single study-based estimates: Describe fully the design features of the single effectiveness study and why the single study was a sufficient source of clinical effectiveness data.	This is minimally addressed, but may be improved by addressing deficiencies in other sections of the manuscript.	No changes requested
	11b	Synthesis-based estimates: Describe fully the methods used for identification	This is minimally addressed. The absence of discussion of the	We have added to the discussion with regards to antimicrobial resistance. Due to lack of any formal evidence on this in this patient group or lack of any current evidence on cost implications of AMR from the literature we felt unable to include these costs

		of included studies and synthesis of clinical effectiveness data.	wider context of the use of the antibiotics in this clinical context prevents a full appreciation of any of the comments that were made.	and benefits as part of an extrapolation of our within trial results.
Measurement and valuation of preference based outcomes	12	If applicable, describe the population and methods used to elicit preferences for outcomes.	This does not seem to have been considered to be part of this study	Quality of life outcomes were measured using the EQ-5D-Y, which is a validated preference based measure and described in the manuscript
Estimating resources and costs	13 a	Single study-based economic evaluation: Describe approaches used to estimate resource use associated with the alternative interventions. Describe primary or secondary research methods for valuing each resource item in terms of its unit cost. Describe any adjustments made to approximate to opportunity costs.	This is not addressed in this manuscript.	We describe data collection for resource use in the subsection titled "Healthcare Use". Identical data collection was used for both the intervention and the comparator. Derivation of the unit costs is described in the subsequent section.
	13 b	Model-based economic evaluation: Describe approaches and data sources used to	This approach was not chosen.	No changes requested

		estimate resource use associated with model health states. Describe primary or secondary research methods for valuing each resource item in terms of its unit cost. Describe any adjustments made to approximate to opportunity costs		
Currency, price date, and conversion	14	Report the dates of the estimated resource quantities and unit costs. Describe methods for adjusting estimated unit costs to the year of reported costs if necessary. Describe methods for converting costs into a common currency base and the exchange rate.	Minimally addressed, although this is not a major deficiency of the manuscript	No changes requested
Choice of model	15	Describe and give reasons for the specific type of decision-analytical model used. Providing a figure to show	Not addressed	No decision-analytical models were used in this manuscript. We have made clearer in the manuscript that we have described quality of life, healthcare use and costs. We have already specified that due to the high rates of missing data, no cost-effectiveness analysis was possible. We have conducted a within trial analysis alongside a trial with a 28 day follow up.

		model structure is strongly recommended.		
Assumptions	16	Describe all structural or other assumptions underpinning the decision-analytical model.	Not addressed	No decision-analytical models were used in this manuscript.
Analytical methods	17	Describe all analytical methods supporting the evaluation. This could include methods for dealing with skewed, missing, or censored data; extrapolation methods; methods for pooling data; approaches to validate or make adjustments (such as half cycle corrections) to a model; and methods for handling population heterogeneity and uncertainty.	1. It appears that the statistical analyses conducted used multiple linear regression with clustering on GP practice. This may be an adequate approach, assuming that any distributional and assumption non-compliance issues were explored during the analytic process. The data being examined are mostly rank ordered in nature, which is conventionally aggregated into scores that are analysed with such linear regression methods. If any doubts about the appropriateness	1. We feel that the linear regression models are appropriate for the data. We generated robust standard errors to account for the clustering within sites. 2. Covariates used were seasonal flu vaccination status (binary variable) and the child's age (continuous variable). The raw values for age were used (i.e. no z-score transformations were performed). These variables were included for the purpose of adjusting for potential baseline differences. As the purpose of our statistical models is to provide reliable treatment effects (as much as possible, given the limitations around missing data), and we have not sought to model the effect of age of flu vaccination status separately, we feel that this approach is appropriate for the purposes of the study.

		s of this usage, it can be checked using a rank-ordered equivalent, such as ordered logistic regression (ologit): if no major difference in interpretation arises from the use of ologit, the results of linear regression analysis are easier for a reader to assimilate. Thus, what is described is in line with my practice in these circumstances. 2. Were the covariate adjustments conducted using raw values, or standardised normal transformations (z-scores)? The way that covariate regression models work is that the constant is adjusted to the mean value of the outcome when all the covariates are zero. This would mean	3. In line with the primary statistical analysis, we have not formally accounted for clustering by sites through hierarchical models due to small cluster sizes. In this paper, we have used robust standard errors which account for any potential additional uncertainty due clustering. We have confirmed that the interpretation of the results are consistent with and without the use of robust standard errors. 4. We have clarified the statement on analysis populations and missing data to state that analyses were performed on an as randomised basis. We then state that available data are used, and have now added that no imputations were performed for missing data. We believe that missing data rates are too high to obtain efficient results from analyses based on multiple imputation. 5. No responses required. 6. We believe that the high rates of missing data, and the potential bias introduced by missing data would make an cost-utility analysis
--	--	---	--

			that raw age adjustment would tell you what the judgement of the child when treated at the time of their birth. The standardised normal transformation ((Value minus mean)/standard deviation) has, by construction, a mean of zero and a standard deviation of 1. If all covariates, excluding the treatment group in an RCT for example, were presented as z-scores, the regression constant would be in the middle of the data distribution (rather than outside the distribution), and all covariates would be corrected for differing scales, having a standard deviation of 1 standard deviation. [I include this comment only for clarification.	inappropriate, and have stated this in the manuscript.
--	--	--	--	---

			Interpretation of numbers is a very fiddly thing, of which details many readers may be unaware.] 3. The comment about clustering on general practices raises the question about whether random effects might have been operating, which might have been addressed by random effects modelling (“mixed” Stata syntax). The conduct and interpretation of the analysis would not be greatly altered by inclusion of this if random effects variable might have been operating and measured. 4. There was a large amount of missing data. No explanation of how this was handled is made in the analytic	
--	--	--	--	--

			methods section, despite the apparent claim that the analysis was performed on an intention-to-treat basis. This should be clarified, and the analysis revised if necessary. 5. The sensitivity analysis, looking at the differential effects in influenza-confirmed and -unconfirmed cases seems appropriate, although the problem of inadequate sample size is acknowledged. 6. It should be noted that no attempt was reported in the manuscript to estimate cost-utility parameters: one method is to plot utility benefits (marginal utility on x-axis) against costs (marginal costs on y-axis), with levels of variability defined using bootstrap	
--	--	--	--	--

			simulation from the trial results. This would make clear the degree of uncertainty that surrounds any judgements being made from a single trial.	
Results				
Study parameters	18	Report the values, ranges, references, and, if used, probability distributions for all parameters. Report reasons or sources for distributions used to represent uncertainty where appropriate. Providing a table to show the input values is strongly recommended	1. It would seem appropriate to have a brief statement about the impact of the randomised treatments on the primary outcome of the trial. This is not included early in the results section. My suspicion is that this arises from the queuing of papers from the trial presented for publication. Has the main paper been published or accepted for publication? This might explain this deficit, not wanting to pre-empt publication of the main results. Otherwise, there is	1. We report the main results from the trial in the background section, with reference to the main clinical paper for the ARCHIE trial. This paper has been published online in March 2021 (https://erj.ersjournals.com/content/early/2021/03/02/13993003.02819-2020) We have updated this reference as it has now been published online.

			no reason not to make a brief comment.	
Incremental costs and outcomes	19	For each intervention, report mean values for the main categories of estimated costs and outcomes of interest, as well as mean differences between the comparator groups. If applicable, report incremental cost-effectiveness ratios.	This is presented for the utility/QoL/symptom score measures in a simple form.	No changes requested
Characterising uncertainty	20a	Single study-based economic evaluation: Describe the effects of sampling uncertainty for the estimated incremental cost and incremental effectiveness parameters, together with the impact of methodological assumptions (such as discount rate, study perspective).	1. The effects of sampling uncertainty on estimated incremental costs were not adequately handled. The costs are predominantly borne by the health system. As such, total costs, and mean cost per patient, are the relevant quantities to be reported. The distribution of these will be skewed, and so estimation of asymmetrical confidence	1. Table 4 reports costs per participant. We have clarified this in the table heading and text. We choose to use adjusted linear regression models to analyse our cost and quality of life data as these models are known to be robust to the violations caused by non-normal data. This has been documented in the literature, e.g. Walters & Campbell, The use of bootstrap methods for analysing health-related quality of life outcomes (particularly the SF-36). Health and Quality of Life Outcomes, 2004. To reassure the reviewer, we have rerun all models using Stata's "bootstrap, bca:" command, and found only small differences between the bootstrapped and non-bootstrapped results. The difference models led to the same interpretation of the data. We have therefore not changed the statistical analysis used in the manuscript. 2. We have clarified that mean costs per participants are presented in Table 4, and that our research focusses on the within-trial

			intervals would be appropriate. This can be achieved by bootstrap multivariate linear regression with estimation of percentile 95% confidence intervals: (“bootstrap, bca:” syntax is placed before the regression model syntax, followed by “estat bootstrap, percentile” to get a table with the percentile 95% CIs, after checking whether this syntax matches your previous syntax). 2. The study perspective used represents an inadequate handling of this particular clinical situation, as commented above.	NHS perspective, limited to a 28-day follow-up.
	20b	Model-based economic evaluation: Describe the effects on the results of uncertainty for all input parameters, and uncertainty related to the		No changes requested

		structure of the model and assumptions.		
Characterising heterogeneity	21	If applicable, report differences in costs, outcomes, or cost-effectiveness that can be explained by variations between subgroups of patients with different baseline characteristics or other observed variability in effects that are not reducible by more information.	It is unclear why the choice of influenza-confirmed is reported in the sensitivity analysis. Other subgroups, such as specific disease related groups might also have been considered, although it is likely that the cases numbers for these subgroups would be too small for confidence in the results. The absence of information about these clinical subgroups in the manuscript prevents a definitive judgement on this question.	We conducted a pre-specified subgroup analysis in participants with laboratory-confirmed influenza because previous data suggesting a possible clinical benefit from early antibiotic use are reported in trials which recruited patients with confirmed influenza (Takeya) or ILI presenting during an influenza epidemic (Maeda), when a high proportion of ILI cases are likely to be due to influenza. Maeda S, Yamada Y, Nakamura H, Maeda T. Efficacy of antibiotics against influenza-like illness in an influenza epidemic. Pediatr Int 1999; 41(3): 274-276. Takeya H, Seki M, Izumikawa K, Kosai K, Morinaga Y, Kurihara S, Nakamura S, Imamura Y, Miyazaki T, Tsukamoto M, Yanagihara K, Tashiro T, Kohno S. Efficacy of combination therapy with oseltamivir phosphate and azithromycin for influenza: a multicenter, open-label, randomized study. PLoS One 2014; 9(3): e91293. We have added this justification to the discussion section.
Discussion				
Study findings, limitations, generalisability, and current knowledge	22	Summarise key study findings and describe how they support the conclusions reached. Discuss limitations and the generalisability of the findings	The previous comments suggest that there has been inadequate discussion of the results of the study. Any conclusions will be of limited applicability	We have added additional discussion points, including on the potential impact of antimicrobial resistance, as suggested by the reviewer. Due to the pragmatic approach of the trial, the study population was highly representative (pragmatic ILI case definition, wide geographical coverage, recruitment from both primary care and other ambulatory care, and cost data available for almost all participants).

		and how the findings fit with current knowledge.	and generalisability. It is possibly/likely, given the low numbers of cases and inadequate economic perspective contained in the data collected as part of the study, that the main value of a manuscript arising from the trial, would be a discussion of how the question of the utility and cost utility of the use of antibiotics (and this specific antibiotic) in influenza-like illnesses in "at-risk" children (however these are defined) might be evaluated in order to assist the health system as a whole to judge when and how those antibiotics should be used. What is provided in the manuscript is, up to this time, inadequate for	We do not believe that the perspective (UK & NHS perspective) was inadequate. Due to the high missing data rates, we have not performed any formal health economics analysis. We have presented available data for EQ-5D-Y, CARIFS, health resource use and costs. We feel that this is an important addition to the literature, as there is a paucity of observational data on this patient population. Due to the limited sample size and the high rates of missing data, these data are insufficient as a basis for clinical decision making, and we have pointed this out in the manuscript, together with recommendations that additional validated quality of life instruments are needed for this patient population.
--	--	---	---	--

			forming those judgements. The authors might usefully consider how such ideas might be rehearsed in a revised manuscript.	
Other				
Source of funding	23	Describe how the study was funded and the role of the funder in the identification, design, conduct, and reporting of the analysis. Describe other non-monetary sources of support.		No changes requested
Conflicts of interest	24	Describe any potential for conflict of interest of study contributors in accordance with journal policy. In the absence of a journal policy, we recommend authors comply with International Committee of Medical Journal Editors recommendations.		No changes requested

VERSION 2 – REVIEW

REVIEWER	Robertson, Iain University of Tasmania, College of Health and Medicine
REVIEW RETURNED	26-Sep-2021

GENERAL COMMENTS	The authors have addressed my previous concerns. Within the limitations of the underlying RCT, some useful information has been obtained. As always with economic analyses, the quantity of data required to answer the economic question (is there value in the use of the treatment in the defined participant group?; is there no value?; or is the value uncertain?) in terms of both costs and benefits are vastly greater than the quantity of data required to estimate the magnitude of benefits in terms of health outcome alone. Both costs and benefits are subject to uncertainty, and this greatly magnifies the sample size requirements to limit that uncertainty to the point that that uncertainty no longer matters. The information in this manuscript represents one step along a very much larger path of data-gathering, which then requires a systemic meta-analysis process to achieve a definitive answer.
--